# Placental Models for Evaluation of Nanocarriers as Drug Delivery Systems for Pregnancy Associated Disorders

**DOI:** 10.3390/biomedicines10050936

**Published:** 2022-04-19

**Authors:** Louise Fliedel, Khair Alhareth, Nathalie Mignet, Thierry Fournier, Karine Andrieux

**Affiliations:** 1Unité de Technologies Chimiques et Biologiques Pour la Santé (UTCBS), Inserm U1267, CNRS UMR8258, Faculté de Pharmacie, Université de Paris Cité, 75006 Paris, France; louise.fliedel@parisdescartes.fr (L.F.); khairallah.alhareth@parisdescartes.fr (K.A.); nathalie.mignet@parisdescartes.fr (N.M.); 2Pathophysiology and Pharmacotoxicology of the Human Placenta, Pre and Postnatal Microbiota Unit (3PHM), Inserm U1139, Faculté de Pharmacie, Université de Paris Cité, 75006 Paris, France; thierry.fournier@parisdescartes.fr

**Keywords:** placenta, experimental models, nanocarriers, pregnancy-associated disorders

## Abstract

Pregnancy-associated disorders affect around 20% of pregnancies each year around the world. The risk associated with pregnancy therapeutic management categorizes pregnant women as “drug orphan” patients. In the last few decades, nanocarriers have demonstrated relevant properties for controlled drug delivery, which have been studied for pregnancy-associated disorders. To develop new drug dosage forms it is mandatory to have access to the right evaluation models to ensure their usage safety and efficacy. This review exposes the various placental-based models suitable for nanocarrier evaluation for pregnancy-associated therapies. We first review the current knowledge about nanocarriers as drug delivery systems and how placenta can be used as an evaluation model. Models are divided into three categories: in vivo, in vitro, and ex vivo placental models. We then examine the recent studies using those models to evaluate nanocarriers behavior towards the placental barrier and which information can be gathered from these results. Finally, we propose a flow chart on the usage and the combination of models regarding the nanocarriers and nanoparticles studied and the intended therapeutic strategy.

## 1. Introduction

Women can experience various diseases during an ongoing pregnancy, just like any non-pregnant women, and will need treatment accordingly [1]. Finding new therapeutics for acute and chronic diseases represents a challenge, especially when it comes to treating pregnant women. It is well established that pregnant women are a sensitive group of patients since they are responsible for their unborn child’s health [2]. During pregnancy many healthcare troubles can arise, such as following the course of a treatment for a chronic condition, e.g., epilepsy [3], or treating an acute disease such as cancer or a pregnancy-associated disorder like preeclampsia [4].

A large majority of women admit to having taken at least one medication during their pregnancy, and current studies report that those numbers are rising [5,6]. Women with chronic diseases are constantly under medication, and healthcare professionals often choose the mother’s benefit over the fetus. Most of the drugs dispensed during pregnancy are prescribed “off-label” because most approved drugs after clinical trials fail to have consistent and appropriate data regarding the use of medication during pregnancy [7]. Even after the thalidomide public health outrage in the early 1960s [8], pregnant women are still considered drug orphans. Pregnant women are not included in clinical trials [9,10]. Consequently, clinicians prescribe medication based on clinical data of non-pregnant women and on pharmacovigilance centers’ teratogenicity knowledge built up through undesirable event occurrence.

Providing safe and effective treatment for pregnant women with a guarantee of their offspring’s safety is a real need. An interesting approach is to develop new formulations based on nanomedicine to better control the behavior of drugs in the mother’s body. Then the safety and efficacy of the nanomedicines must be studied in the context of the pregnancy. The placenta is a key feature to evaluate the communication between the mother and the fetus. Better understanding of this biological interface and how it interacts with drugs and nanocarriers could help with pregnant women’s treatment management.

Nanocarriers as drug delivery systems for pregnancy care are considered promising candidates to answer these peculiar specifications. They are known to improve pharmacological properties of conventional drugs such as poor water solubility, low bioavailability, biological degradation, first-past metabolism, and side effects [11]. Therefore, well-designed nanocarriers could diminish the risk of affecting the fetus by preventing any transplacental passage, and at the same time could ensure an efficient treatment to the mother or vice-versa or to specifically target the placenta for placental diseases.

In this review, we analyzed published articles reporting the evaluation of nanocarriers as drug delivery systems for pregnancy-associated disorders. Therefore, studies discussing the toxicity or impact of environmental particles on pregnancy outcomes were not included in this review. The aim of the current study is to review the various models that could be used to evaluate nanocarrier interaction with the placental barrier for novel therapeutics development for pregnant women. For easier understanding, nanoparticles referring to model nanoparticles used for fundamental studies are designated as nanoparticles (NPs), and nanoparticles studied as drug delivery systems are named nanocarriers.

In this review, will first present the rising interest in nanomedicine for pregnancy-associated disorders. Then the diversity of models developed to study the interaction of nanocarriers with the placenta will be discussed. They can be divided into three common categories: in vivo, in vitro, and ex vivo models. The models of each category are detailed in the following tables. An analysis of scientific publications describing the use of those models is performed to highlight the most pertinent methodologies to study nanocarriers as drug delivery systems in the context of pregnancy. Finally, among the scientific publications analyzed, we highlight the absence of a consensus on which model of the placental barrier to use to evaluate a specific property of a nanocarrier. Therefore, we propose a flow chart on how to use placenta-based models to screen and to help to build a homogenized work package to evaluate nanocarrier potential for pregnancy-associated disorders.

## 2. Nanocarriers for Pregnancy-Associated Disorders

It is not possible to translate drug clinical data from a non-pregnant woman to an expecting mother since women’s bodies experience various changes during pregnancy [12]. Almost all physiological functions in a woman’s body is adapted to allow for the fetus’ development [13], which will affect the drug disposition. Absorption of the drug is modified for pregnant women because of the delayed gastric emptying, decrease of gastric pH, and intestinal motility. The distribution of hydrophilic drugs is changed through the increase in the total body water, and the distribution of hydrophobic drugs is increased with a higher fat compartment. The modification of CYP450 and UGT activities affects the metabolization of xenobiotics. Finally, the renal clearance of most of the drugs is increased thanks to the increase in cardiac output [14,15].

All these body modifications in pregnant women can change the benefit/risk ratio of a drug for the future mother. However, they can also influence the risk for the fetus by increasing the concentration of drugs in contact with the placenta and favor its transplacental passage. During pregnancy, the principal route of communication between the mother and the baby is governed by the placenta. Therefore, fetal exposure to any substance depends on the transplacental passage.

The placenta can be described as a utero–fetal structure since it comes from maternal and embryonic cells. The human placenta is a hemochorial transient organ, which means that the mother’s blood is in direct contact with the placental cells. As presented Figure 1, the placenta is attached to the mother’s uterine wall, and the blood is discharged into what is called the intervillous space from the spiral arteries. The fetus is linked to the placenta by the umbilical cord, which groups two fetal arteries and one vein. The blood circulation is then divided into arborescent structures called the chorionic villi. These villous structures are composed of three different cell layers: the syncytiotrophoblast, the villous cytotrophoblast, and endothelial cells. The syncytiotrophoblast is derived from the fusion of cytotrophoblasts to form large syncytia called the syncytiotrophoblast. This cell layer forms the outer part of the villi and is in direct contact with the maternal blood, ensuring the transplacental passage of nutrients and gases. The placenta provides all the needed substances to ensure the healthy development of the baby.

Even if the placenta is designated as a barrier, it cannot prevent the passage of all exogenous components that are administered to the mother. Nowadays, many common medications are small molecules, which can cross from the maternal side to the fetal side. Today, gathering as much knowledge as possible about drug safety, teratogenicity, and efficacy in the context of pregnancy is a necessity for healthcare professionals to deliver the best medical care [16,17]. Modulating the transplacental passage of drugs can be achieved thanks to the advancements in pharmaceutical technology and drug delivery. Therefore, nanocarriers as drug delivery systems could address this specific challenge for women’s treatment during pregnancy.

Nanocarriers are defined as nanoparticles with a size from 1 to 100 nm [18]. With specific drug delivery systems, it is possible to enhance pharmacokinetic properties of the drug by increasing the residence time and avoiding the rapid clearance of the compound. Coating the surface of nanocarriers with polyethylene glycol (PEG) chains of various lengths, called PEGylation, is a common technique used to improve systemic circulation time and decrease immunogenicity. This confers stealth properties to the administered nanocarriers [19]. The administered PEGylated nanocarriers will circulate long enough to deliver the active pharmaceutical ingredient (API) to the desired target by passive targeting.

This passive targeting relies on the carrier characteristics (size, surface charge, stealthiness) and the biological environment, but does not allow it to enter a specific organ or specific cell type. A specific ligand can be grafted to the nanocarrier and therefore bind the desired target by active targeting or vectorization [20].

Choosing the physicochemical and surface properties of nanocarriers allows the biodistribution of the drug to be controlled and modified [21,22] and the physiological changes during pregnancy to be coped with. The use of nanomedicine is a promising approach to lowering toxicity, increasing therapeutic efficiency, and having a controlled and sustained released of the API [23,24,25,26] and thus for pregnant women’s treatment [27].

The rising interest in the development of nanocarrier-based medicine for pregnant women begs the question of their evaluation and safety considerations. Clinical trials including pregnant women are restricted and therefore imply proper preclinical evaluation models. Human placenta is donated from hospitals and has several characteristics, and it is defined with high inter-species differences compared to other mammals. The use of human cells and tissues is key to avoid variation between species to ensure complete and trustworthy results [28,29].

## 3. Which Placental Evaluation Models to Evaluate Nanocarriers

In cellular evaluation models are used to study the fate, the potential toxicity, and the efficacy of newly developed medication. They can be divided into three categories: In vivo, using living animal models;In vitro, using one or several cell types in culture;Ex vivo, using mostly organs or parts of organs outside of a living organism.

Placental in vitro and ex vivo evaluation models arise from human origins. For the evaluation of nanocarriers, models should be adapted and compared to other models for the evaluation of conventional dosage forms. 

### 3.1. In Vivo Models 

Animal models are live organisms and therefore have the advantage of exhibiting several properties useful to studying the future of administered medicines and their possible toxic effects on the body. They indeed give useful information specifically about learning about the biodistribution of molecules in a living organism. This knowledge has a special importance in the development of nanomedicines. Drug loading into nanocarriers aims to modify its distribution or to achieve its targeting to a specific organ. In vivo experiments are usually performed to demonstrate the proof of concept for the strategy of nanomedicines, i.e., decreased toxicity and increased efficacy compared to the conventional dosage form.

All the analyzed studies using in vivo placental models to study nanocarriers are grouped in Table A1.

Many experiments on pregnant animal models have been performed to study and understand the interaction with the placental barrier of standardized inorganic nanoparticles, e.g., gold [30,31,32,33], polystyrene [34,35], silica nanoparticles [36,37,38,39], quantum dots [40,41], and carbon nanotubes [42] with calibrated size and shape. These studies commonly suggest that the transplacental passage in vivo in murine animals depends highly on the size, the nature of the coating, and the gestational maturity of the placenta. Nanoparticles and nanocarriers under 100 nm exhibit a higher transplacental passage than other NPs at every stage of the placenta’s development. It seems that feto/placental accumulation of NPs is gestational-stage dependent, i.e., the transplacental passage of NPs can be facilitated at an early gestational age for numerous NPs of diverse sizes compared to a late pregnancy stage [32,33,39,43]. A specific coating could prevent transplacental passage depending on the nature of the NP, such as, for example, PEG-coated QDs [40]. Not all NPs able to cross the placenta and reach fetuses always have harmful effects on the offspring and the mother, and the toxic effect could be attributed to certain physicochemical characteristics as a small size and positive surface charge [36,38].

Other teams used healthy pregnant animals to evaluate nanocarriers as drug delivery systems, such as liposomes to deliver higher concentrations of drugs to the placenta such as indomethacin for the treatment of preterm labor management [44,45] or vasodilators to treat fetal growth restriction induced by impaired uteroplacental blood flow [46] or to ensure that gadolinium, a contrasting agent for medical imaging of the mother, does not cross the placenta [47]. These studies concluded that liposomes are suitable drug delivery systems to control biodistribution compared to the free drug because they modify the physicochemical properties of the liposomes.

Changes in biodistribution can be improved by modifying the surface of nanocarriers with PEG chains or by adding a specific peptide or antibody to target a specific receptor present at the surface of the placenta [48] and encouraging results have arisen from these studies. King et al. designed a tumor-homing peptide and RGD-coated nanocarriers, which exhibited efficient targeting towards the placenta in gravid mice compared to non-functionalized nanocarriers and fetal transfer. Then they tested insulin-like growth factor-loaded nanocarriers in a pregnant mouse model of fetal growth restriction and showed an improvement in fetal and placental weight compared to the free drug [49]. Specific placental targeting was performed by using functionalized liposomes with a peptide derived from VAR2CSA. This protein is known to be produced by plasmodium falciparum-infected erythrocytes in a context of pregnancy malaria. VAR2CSA specifically recognized the chondroitin sulfate A at the surface of the placenta and facilitated the penetration of the parasite into the placenta. Zhang et al. based the functionalization of the liposomes on this mechanism and demonstrated a significant translocation of liposomes inside the placenta compared to the other organs. In the context of choriocarcinoma, the targeted delivery of methotrexate to the trophoblast cells showed high therapeutic efficiency compared to free methotrexate and lower toxicity [50]. 

Certain studies were performed on pathological animal models to evaluate the therapeutic efficiency of nanocarriers in the context of pregnancy-associated disorders such as uterine inflammation [51,52] or preeclampsia [53,54]. Some nanoparticles/nanocarriers were experimented with in a gravid mice pathological model of intrauterine inflammation. Tian et al. demonstrated that the intrauterine inflammation could enhance maternofetal transfer of NPs during a late stage of gestation. Injection of gold nanoparticles of 3, 13, and 32 nm in a pathological model of pregnant mice showed an increased accumulation of NPs of 3 and 13 nm inside fetal tissues compared to healthy mice. The 32 nm NPs could not cross the placental barrier in either model [51]. Another study focused on testing the therapeutic efficiency of dendrimer NPs containing N-acetyl-L-cysteine (DNAC) [52]. The NPs were administered via intraperitoneal injection to the pregnant mouse with intrauterine inflammation. The results demonstrated that DNAC significantly reduced the preterm birth rate and altered the placental immune profile with a decrease of CD8+ cell infiltration, an improvement in neurobehavior, and a reduction in the neuroinflammation of fetuses, reflecting the therapeutic efficiency of the administered drug. 

One of the physiopathological mechanisms of preeclampsia (PE) is characterized by an overexpression of an antiangiogenic factor, sFlt1, provoking placental ischemia and the secretion of inflammatory and oxidant factors. This induces vascular endothelium damage in the mother (systemic vasculopathy) with vital organ impairment, leading to potential eclampsia. Direct consequences are the termination of pregnancy and placenta expulsion, endangering the mother’s and the baby’s life. Many potential therapeutics developed are based on the downregulation of sFlt-1. Yu et al. used a pregnant rat model of preeclampsia by injection of TNF-α to test PAMAM nanocarriers loaded with an anti-sFlt1 siRNA. In vivo results showed a decrease in circulating levels of sFlt-1, mean arterial pressure, and a significant increase in fetuses’ weight and placentae compared to the control groups. This suggests that siRNA-sFlt1-PAMAM has a positive effect on preeclampsia symptoms in the PE gravid rat model [53]. 

In another study, a new preeclampsia pregnant mouse model was developed based on placental RGD-targeting liposomes to deliver a specific siRNA to induce preeclampsia-like symptoms. Formulations were tested on an in vivo model of healthy pregnant mice, and a biodistribution study showed a higher uptake of RGD liposomes than regular PEG liposomes. After collecting the mice’s placenta and pups, results showed an increase in sFLT-1 mRNA expression by the trophoblasts. [54]. 

Few studies testing nanocarriers on pathological in vivo models have been reported. One hypothesis to explain this underuse of pathological animal models could be the lack of knowledge of the pathophysiology mechanisms of diseases in humans. It is certainly easy to reproduce symptoms but difficult to reproduce the physiological mechanisms, and therefore difficult to obtain a reliable animal model. For example, pathological models used for preeclampsia can arise from genetic modifications as the STOX-1 model [55], from surgical procedures as the reduced uterine perfusion pressure (RUPP) model [56,57,58], or from drug induction with TNF-α [53]. Such models can recreate the symptoms but are missing the accurate mechanisms of the disease and therefore are not completely trustworthy to test therapeutic efficiency and cannot be uniquely considered to screen future treatments [59]. 

Rodents exhibit anatomical and physiological placenta characteristics close to humans, compared to other species (excluding non-human primates), as presented Table 1 [60,61,62].

They appeared to be the ideal in vivo model to use for biomedical studies because of their small size, ease of maintenance, and short life cycle [63]. However, the most important part is to understand that an in vivo model must be chosen regarding the pathology and the API studied. Regarding the study of nanocarriers as drug delivery systems, mice and rats have proven to be useful to evaluate their biodistribution, placental targeting, transplacental passage, fetotoxicity, and therapeutic efficiency on specific diseases. Despite these studies and even with the similarities in placental structure and physiology between rodents and humans, there are still some differences between those species [64]. Other characteristics should also be taken into account when choosing an in vivo model to evaluate nanomedicine in the context of fetal medicine and obstetric studies, such as the placentation mechanism, the number of fetuses per pregnancy, and the gestation length [65]. Therefore, other species could be encountered in this type of trial, like rabbits or sheep [66,67].

To study human medical matter will require, at some point, human material such as cells to overcome species differences. Therefore, in vitro models can bring useful answers to nanocarrier evaluation. 

### 3.2. In Vitro Models

#### 3.2.1. Cell Culture 

In vitro models are widely used to study different functions of the placental barrier and the interaction of exogenous molecules and nanocarriers [28]. These cell lines are derived from human cells. They can be divided into two main categories: immortalized cell lines and primary human trophoblasts.

Placental cell lines originate either from a patient’s choriocarcinoma, which are already immortalized cancerous cells, or from trophoblasts, which are freshly isolated from a unique patient’s placenta and immortalized with an oncogene virus [68]. Either way, each category comes from the same origin and ensures minimum variability in between experiments. Since these cells are cancerous, they exhibit the key parameter to be easily maintained in culture [69]. Thanks to their defined protocols and optimized culture conditions, cell lines are easy to use and allow reproductive experiments to be performed [70].

Using cell lines provides an in vitro alternative to obtain quantitative and qualitative assessment of NP transfer across the placental barrier [2,71]. The different categories of placental cell lines available to evaluate nanocarriers are summarized the Table 2. Among them, the most common cell lines used are choriocarcinoma cell lines, which are cancer cells isolated from a patient’s uterine tumor, such as BeWo, JAR, and JEG-3 cell lines [72]. On the other hand, HTR-8/SVneo [73] and Swan 71 cells [74] are healthy trophoblasts immortalized by the transfection of oncogenes. Finally, a new kind of cell emerged, the ACH-3P cell line, which is a combination of primary human first trimester trophoblasts with a human choriocarcinoma cell line (AC1-1, a mutant from the JEG-3 cell line) [75]. Those cell lines exhibit close characteristics to the primary trophoblasts [76].

These cell lines often are used to investigate the transplacental passage of environmental nanoparticles and therefore their potential toxicity on the fetus [77]. Some studies focused on the evaluation of the nanomedicines’ interaction with the placenta to treat the mother, the fetus, or the placenta.

All studies using in vitro placental models to study nanocarriers are grouped in Table A2.

##### BeWo Cell Line

BeWo cells were produced by Patillo and Gey in the 60s [78]. Those cells were isolated from a patient choriocarcinoma, which was transplanted in series inside a hamster’s cheek. It was defined in the 1990s that the BeWo cell line corresponds to undifferentiated cytotrophoblast-like cells and possibly giant multinuclear cells that resemble the syncytiotrophoblast [79,80]. This cell line can easily form confluent and polarized cell monolayers. They have a similar morphology and exhibit many characteristics of the third trimester trophoblasts. They do not spontaneously form a syncytium, but it is possible to induce syncytialization with a treatment with forskolin or cycling adenosine monophosphate [81]. These cells secrete cytokines and hormones and express cytokeratine-7, which is a characteristic trophoblastic/epithelial marker [82,83]. BeWo cells can be used to understand physiological mechanisms linked to placentation, such as villous trophoblast fusion [84], the syncytialization process, or adhesion, endocrine, or metabolic function [85], but also to understand infection pathways of parasites like *Toxoplasma gondii* [86,87] or viruses like HIV [88]. BeWo cells exhibit most of the key properties of the real physiological placental barrier, which is why it is considered a satisfying in vitro model [89].

This cell line is the most used and has been implanted in scientific research for several decades. This model appears to be suitable to study NP interaction with the placental barrier [24,90], as can be seen in the many studies performed.

BeWo cells are often used to evaluate the toxicity of inorganic nanoparticles such as iron or silica nanoparticles to the fetus. Poulsen et al. used the BeWo b30 cell line and evidenced the accumulation of silica nanoparticles in the cells and no toxic effect under a concentration of 100 µg/mL [91]. Alpha-Fe2O3 NPs of 15, 50, and 78 nm incubated with BeWo cells demonstrated that the largest NPs (50 and 78 nm) disrupted the epithelial barrier integrity of the BeWo monolayer [92].

This in vitro model can also provide an evaluation of drug delivery through nanoparticulate systems to treat diseases affecting the mother without limited risk of potential transplacental passage.

Doxorubicin is a chemotherapy drug that is often used against breast carcinoma. Nowadays it is contraindicated to inoculate chemotherapy for a woman when she develops cancer during pregnancy due to its passage into the fetus. She often must wait until the end of the pregnancy or stop the pregnancy to be treated. Nanocarriers have been proposed as a strategy to avoid the fetal toxicity of doxorubicin. Soininen et al. investigated the penetration of doxorubicin-free or loaded into liposomes with the BeWo cell line. The results showed that PEGylated liposomal doxorubicin has a lower uptake and therefore toxicity on the cells compared to free doxorubicin and unPEGylated liposomal doxorubicin formulation [93]. Sezgin-Bayindir et al. studied the placental transport, and the cytotoxicity of clonazepam “micelle-like nanoparticles” (MNP) exhibited a cellular toxicity on BeWo compared to the free drug. [94].

This in vitro model of the cell line helped to understand the behavior of nanomedicines towards placental cells and demonstrate the efficiency of drug delivery using NPs. Albekairi et al. studied the transport of digoxin-loaded PEGylated polymeric nanocarriers. Digoxin is indicated in case of cardiac failure and atrial fibrillation in adults and for fetal arrhythmias. During fetal development the outbreak of arrhythmia can lead to fetal congestive heart failure and hydrops fetalis. To avoid those consequences, it would be mandatory to administer the medication directly to the fetus. The key point in this study is to deliver the drug only to the fetus and not to the mother. Experiments on the BeWo cell line displayed a higher penetration of digoxin-loaded nanocarriers into placental cells than the free digoxin [95]. Other polymeric NPs loaded with dexamethasone were designed to treat fetal congenital adrenal hyperplasia. In vitro assays highlighted the potential of those NPs to cross the BeWo cell layer and therefore to potentially reach the fetus in vivo [96].

In comparison to monolayer seeded BeWo cells, it is possible to grow them on Transwell^®^ inserts to investigate the transplacental passage of the nanocarriers, as presented in Figure 2. Transwell^®^ devices exhibit two compartments separated by a cellular layer in conditions closer to the physiological ones [97]. In the case of the placenta studies, it recreates the maternal compartment in which drugs are incubated and the fetal compartment, both separated by the BeWo cells (which can be combined with other cell types) that mimic the placenta.

It is also shown that changes in NP surface exhibits a key role in the translocation process, i.e., two NPs of polystyrene core but with different coatings, such as fluorescent polystyrene nanoparticles covered with carboxylate or amine groups, exhibit different translocation rates [97,98]. Correia et al. incubated iron dioxide and silica nanoparticles with sizes of 8, 25, and 50 nm on BeWo cell monolayers seeded on Transwell^®^ inserts. Results showed that the toxic effect, uptake, and transport of both NPs were influenced by the surface coatings, which may have been due to a change in surface charge, but no impact linked to the size or concentration of the NPs was observed and no NPs were observed inside the fetal compartment [71]. The crossing of nanocarriers through BeWo cells on Transwell^®^ inserts also showed that pullulan nanoparticles with higher sizes (200–300 nm) were translocated up to the basolateral compartment of placental cells (which correspond to the fetal side) and thus suggests that those bigger nanoparticles could more easily reach the fetus [99].

The usage of BeWo cells to mimic the placental barrier in vitro has undoubtedly proven to be a model of choice to evaluate the behavior of such nanoparticles and nanocarriers towards a biological barrier. However, its single use does not seem to be sufficient to provide reassuring data regarding the consequence of administrating nanocarriers during pregnancy. As a matter of fact, many of the studies using BeWo cell lines added some experimentation using ex vivo models such as perfused human placenta [89] and human placental explants [93,100].

##### JEG-3 Cell Line

JEG-3 cells are derived from choriocarcinoma. They exhibit interesting endocrine function by producing progesterone, human chorionic gonadotrophin (HCG) [101], steroids, placental hormones, and enzymes [102,103]. They have properties close to human placental villous trophoblasts, such as a mechanism of proliferation, invasion of syncytiotrophoblast, and cell differentiation [104]. These cells also exhibit some potent microbial resistance to *Toxoplasma gondii* infection like BeWo cells [105].

These characteristics have been utilized to determine the inhibitory effect of magnetic NPs coated with Fe3O4-dextran-anti-β-HCG containing heparinase antisense oligodeoxynucleotide (ODN) on choriocarcinoma tumor growth. Evaluation of the invasion and proliferation characteristics and heparinase expression of the JEG-3 cells highlighted that NPs could deliver the ODN to the choriocarcinoma cells and sufficiently inhibit the invasion and proliferation of the JEG-3 cells. Moreover, it demonstrated the potent use of the JEG-3 cells as an effective in vitro model for NPs evaluation in a context of choriocarcinoma [106].

Similarly, Zhang et al. also evaluated NPs to treat choriocarcinoma tumor growth by using JEG-3 cells. They formulated specific NPs coated with chondroitin sulfate A-binding protein (CSA-BP NPs), which is derived from VAR2CSA and loaded with doxorubicin. It was observed that the NPs were efficiently internalized by the lysosomes in the JEG-3 cells and the anti-tumor activity was increased [107]. 

These works evidenced that JEG-3 cells are useful to predict the cellular penetration of NPs and the efficacy of drugs delivered inside this model derived from choriocarcinoma. Only anticancer activities of drug-loaded NPs have been published with these cells.

##### HTR8/SVneo

HTR-8/SVneo cells are the extra villous trophoblast (EVT) cell line of reference [108]. They exhibit the functional and molecular characteristics of first trimester extra villous trophoblast (EVT) cells in the pregnancy context [109,110,111,112].

These cells showed some versatile utilization, such as congenital diagnosis development, proteomic data acquisition, and NPs as drug delivery system evaluation.

HTR-8/SVneo have been investigated to improve a diagnosis test based on the sampling of fetal DNA in trophoblast chorionic villi to search for any congenital disorders [113]. With this aim, magnetic NPs coated with specific leucocyte antigens have been designed to isolate the EVT to retrieve the fetal antigen in a non-invasive way [114]. 

Finally, the HTR-8/SVneo cell line was used to evaluate the siRNA transfection using PAMAM NPs in a preeclampsia context. Results showed high cellular intake, significant decrease in sFlt1 secretion, and an effect on cell proliferation, concluding in promising results to treat preeclampsia [53]. Yu et al. also used the HTR-8/SVneo in vitro model to evaluate the transfection efficiency of siRNA-loaded liposomes grafted with RGD to target the placenta to induce PE symptoms in a mouse model. Results could highlight the efficiency of RGD liposome cellular uptake enhancement and the endosomal escape of siRNA inside cells. These results confirmed the in vivo experiment and were in favor of an efficient strategy to induce PE-like symptoms in mice compared to the control [54].

##### Comparative Nanoparticle Translocation Study between BeWo, JEG-3, JAR, and ACH-3P Cell Lines

A comparative study aimed to understand the differences between four in vitro placental cell lines [68]. It compared three choriocarcinoma cell lines: BeWo, JEG-3, and JAR cell lines and the first trimester trophoblast hybrid cell line ACH-3P. The authors evaluated five parameters to see whether each of those cell lines could be used as a relevant in vitro placental barrier model: the evaluation of the tight junction setup, the transepithelial resistance (TEER), the glucose transport, the hormone secretion, and the transplacental passage of polystyrene nanoparticles. Regarding the interaction with the nanoparticles, it appears that the nanoparticle transport is very dependent on the in vitro model chosen. The results illustrated that ACH-3P and JEG-3 are more permeable to 50 nm and 490 nm polystyrene nanoparticles than the JAR and BeWo. In addition, all cell types exhibited a higher apical-to-basal transport ratio for 50 nm nanoparticles than the 490 nm ones. In comparison between cell lines, BeWo and Jar formed a cellular barrier that 50 nm nanoparticles could not easily pass compared to ACH-3P and Jeg-3. They also compared these results to those of ex vivo experimentation with a perfused human placenta. Only ACH-3P had similar results as the ex vivo model for 50 and 490 nm. Finally, they concluded that ACH-3P and JEG-3 exhibit closer behavior toward nanoparticles transported to the physiological placental barrier than the BeWo and JAR cell lines, since BeWo and JAR had a low transfer for nanoparticles under 100 nm.

Cell lines are very useful in terms of experimental conditions and in reproducibility of the results. However, they also lack many functions and parameters that allow them to be as close as possible to the physiological conditions. Unlike human primary trophoblasts, placenta-like cell lines have many differences compared to the in vivo state, such as their morphology, secretion of hormones, cellular trafficking, etc., because of their genetic modifications, which can alter their physiological properties [28]. For this reason, they are better considered as an easy and quick way to have access to basic information.

##### Primary Human Trophoblast Cell Culture

Among in vitro placental models, primary cell cultures of human trophoblasts have a significant place. They are obtained by several consecutive digestions of human placenta tissue; trophoblastic cells are subsequently isolated and cultured [115,116,117]. 

The team of Juch et al. got interested in a more fundamental type of nanoparticles, which are fluorescent polystyrene nanoparticles. Incubation with primary human trophoblasts showed that the surface charge of the nanoparticle plays an important part in the way the particles interact with the placental cells, with a higher uptake of the positively charged nanoparticles [100].

Bajoria et al. performed the culture of purified primary human trophoblasts to evaluate the uptake pathways of small unilamellar liposomes of different charges encapsulating a fluorescent compound, the carboxyfluorescein. The results showed that carboxyfluorecein encapsulated inside cationic liposomes was as internalized as free carboxyfluorescein, whereas carboxyfluorescein inside neutral and anionic liposomes was more uptaken. These results illustrated the influence of nanocarrier surface charge on hydrophilic small molecule internalization by trophoblasts. In should be noticed in this study that only the encapsulated molecules were followed, not the nanocarrier, which gives indirect information about liposome uptake by cells [118].

Another study performed by our team evidenced the delivery of small interfering RNAs to the primary human villous cytotrophoblast placental cells by the help of three different lipoplex formulations containing different cationic lipids (DMAPAP, DDSTU, or CSL-3) stabilized by an anionic polymer to neutralize the overall surface charges of the nanocarriers. Lipoplexes were able to deliver siRNA inside the syncytiotrophoblast, and results showed the potential of those formulations, especially the one containing DMAPAP, to deliver siRNA to the placenta without potential transplacental passage [119].

Among the benefits of primary trophoblasts as a placenta model, their high biological relevance makes them the closest to human physiologic conditions. In addition, the isolation of primary trophoblastic cells from one placenta at a time allows for the real-life inter-variability between patients. This model also avoids the genomic trouble seen in cell lines. Because of this, they can give lifelike information when used in research studies. Despite the existence of such benefits, primary human trophoblasts exhibit some significant drawbacks. First, their limited lifespan and their slowness to grow can cause some difficulties in carrying out research experiments. This type of model needs optimized and accustomed culture conditions and protocols, which increase the uncertainty of success and imply additional workload to the study. In addition, there is a need for secured and qualified structures and equipment, and trained laboratory staff. Finally, one of the major downsides of using human primary trophoblasts is the way to obtain the cells. for this purpose, researchers need to have access to freshly delivered placenta, to which patients must give their written consent, which means involving an ethical committee in the process [120,121]. 

This model can also be combined with cell lines. As a matter of fact, cell lines are often easier to handle than primary cell cultures but lack some important features compared to physiological conditions [122,123]. In addition, one important difference between primary human villous trophoblasts and immortalized trophoblast cells is the inter-individual variability of the cell nature. Commercial cell lines avoid the possible interference in the results due to this variability brought by using different placentas in the same study [124]. 

About in vitro models, it must be noted that commercial cell lines do not exhibit all the necessary characteristics to precisely predict the fate of nanomedicines in vivo, primary cell culture demands heavy and difficult setup protocols (short lifespan, difficult to isolate the right cells, etc.) and are fragile and sensitive to being handled. Finally, the major limit here resides in the 2D structure of the cell cultures. Indeed, cells are plated in Petri dishes or wells as a single cell layer, and this is missing the fact that the placenta constitutes multiple cell types and layers. The placenta is also a dynamic system that is not reproduced in the conventional cell culture models.

#### 3.2.2. Emerging In Vitro Models

New innovative in vitro models have been developed in the last decades to fit better to human physiological conditions. By engineering three-dimensional cell models that can recreate a micro tissue environment, it seems possible to get closer to the physiological situation. Those new models can be divided based on their methods of manufacturing into three subgroups: (i) co-culture based on the association of layers of different cell types to recreate a close environment to in vivo situation, (ii) 3D cell cultures or organoids recreating self-renewing 3D systems copying the organs’ organization in vivo with various cell types, and (iii) placenta-on-a-chip combining co-culture methods and microfluidics technology.

Drwal et al. developed a co-culture of JEG-3, BeWo, syncitialized BeWo, and adrenal cells to get closer to the real interactions and metabolic behavior of cells inside the fetoplacental unit. Thanks to this method, they could exhibit the diverse hormone secretion of the different cell lines and use these differences to show a complementarity between the different cells and create a more complete cellular model [125]. No experiment has yet been performed with nanoparticles or nanocarriers on this promising model.

The transplacental passage of nanoparticles has been examined by another type of co-culture combining layers of trophoblasts (BeWo) and placental endothelial cells (HPEC-A2) to mimic the presence of blood vessels inside the culture. This model allowed researchers to demonstrate that each type of cell had similar retention capacity towards the incubated polystyrene nanoparticles [90].

BeWo cells and human placenta-derived pericytes have also been associated with co-culture on Transwell^®^ membranes to recreate a blood–placenta barrier using trophoblast-like cells and to recreate a microvascular environment with the pericytes. The medium in contact with the BeWo cells represents the maternal compartment and the medium in contact with the pericytes is considered the fetal compartment. The interaction of magnetic nanoparticles exhibiting different charges was studied using this co-culture model and showed that the cationic particles had a higher interaction and were retained by the cells, whereas the anionic and neutral particles could easily cross the barrier and be found in the fetal compartment after 24 h [126].

Taken together, co-culture models appear to be a great improvement concerning in vitro cell models, since they exhibit cell line advantages with more refined specificities, even if the setup and validation are still considered limitations for these complex models in terms of studying new therapeutics.

Some co-cultures exist as three-dimensional setups, which are miniature organizations of cytotrophoblast, extra-villous cytotrophoblasts, and syncytiotrophoblasts cultured together that mimic the physiological disposition and functions of an entire placenta in its early stage of the first trimester [127,128,129]. One of the major properties of such a model is that it can self-renew and therefore be kept for a long time in culture [130].

Muoth et al. developed a 3D co-culture model of placenta formed of placental fibroblasts covered by a layer of trophoblastic cells. With this structure, they demonstrated getting closer to human in vivo hormone secretion, studied the course of nanoparticles inside the feto-placental unit, and obtained acute toxicity data [127]. The team assessed the penetration of gold nanoparticles inside the placental cells by using this type of 3D co-culture microtissue of placenta [131]. They found that the cellular uptake was higher and deeper for small or sodium carboxylate-modified NPs than for larger or PEG-modified NPs.

Trophoblast organoids are another type of primary trophoblast in vitro model. This model originated from the isolation of trophoblastic cells and is cultured to help the cells differentiate into both syncytiotrophoblast and extra-villous cytotrophoblasts. Usually, when primary trophoblasts isolated from sampled human placenta are cultured in a Petri dish, they rapidly lose their ability to proliferate. Here, Turco et al. isolated stem cells from human placenta and cultivated them with growth factors [128].

A “placenta-on-a-chip” model has been described and consists of a microfluidic chip comprising BeWo cells and human primary placental villous endothelial cells cultured on matrigel to allow cell adherence under flow conditions, as presented in Figure 3. [132]. This chip undergoes a specific flow conferring a dynamic component compared to the previous in vitro placental models. It aims to mimic all the structural and functional complexity of this organ in an in vitro device. This technology allows the trophoblast cells to fuse and change into syncytiotrophoblast.

This newly developed device has already been investigated in several studies on various topics, such as the behavior of the placenta towards bacterial infection [133] or the transplacental passage of drugs such as gestational diabetes drugs [134] or caffeine [135]. Finally, the exposure of environmental nanoparticles such as titanium dioxide nanoparticles using this micro-engineered device revealed that cells suffered from dramatic physiological changes [136].

As these models are quite recent, there are few studies on the interaction of nanoparticles with the placental barrier using organoids. They can grow slowly by themselves and appear to be relevant models to understand the defects of placenta involved in several pregnancy disorders. They seem to be the closest in vitro model to real-life conditions. They could give useful information on the behavior of the placental cells when in contact with exogenous nanoparticles on a molecular and cellular level.

Despite all that, those emerging in vitro models still encounter some difficulties in being designed and reproducible and are not yet considered unique and reference models to study the fate of nanomedicines in a pregnancy context [137].

### 3.3. Ex Vivo Models 

Ex vivo models allow the use of complex tissue organization originating from an organism without completely changing its natural environment, i.e., recreating an in vivo situation at the bench [138,139]. In opposition to the huge difficulties of designing 3D cell culture models, ex vivo models offer direct access to intact 3D structures integrating all physiological constituents. Ex vivo models can be obtained through the dissection of the organ or part of it without compromising its physiological integrity. In addition, ex vivo models provide useful, transposable, close-to-reality, and diversified information compared to the in vitro and in vivo models described before.

In the case of placenta, access to human tissue is a real advantage, especially when studying a physiological barrier that requires 3D experimental models containing exhaustively all cell types. Human placenta can be collected with the patient’s consent after birth if the hospital is involved in clinical research, unlike other barriers such as the blood–brain barrier, for which it is not possible to access to the required tissue easily or ethically. Due to the species specificity of placenta, human placenta-based models are the closest ones to the physiological situation. As these models rely on a single patient’s placenta, they provide inter-variability in between patients, which can be found in real-life situations.

The most impairing disadvantage of placenta-based ex vivo models is the short time to keep them intact in culture. These models can be maintained in culture for less time than human trophoblast primary cell culture. From another side, they have the same drawbacks as the primary cell culture, which is the need to have access to human tissue from close maternities and the need to have trained personnel and specifically equipped structures.

To study the human placenta, two types of ex vivo models exist: the dually perfused placenta and the placental explants. All studies using ex vivo placental models to study nanocarriers are grouped in Table A3.

#### 3.3.1. Perfused Human Placenta

The dually perfused placenta consists of sampling an entire placenta after C-section and keeping it alive for several hours. The model was designed by Panigel et al. and optimized by Schneider et al. [140,141]. The purpose of this model is to recreate the materno-fetal blood circulation by placing catheters in the spiral arteries corresponding to the mother’s circulation and in the embryonic arteries leading to the fetal side. 

Fetal and maternal circulation is re-established by a pump system, as shown in Figure 4, and the transport of a chosen test substance can be investigated. This perfusion model is a simplified model of placental transport, and it does not take all the physiological and biochemical variables in the mother and fetus into account. The model can only represent transport in the late third trimester. However, the asset of the model is that the in vivo placental metabolism is still intact and the assessment of substance binding to placental tissue can be investigated.

The setup is the most delicate part of this model. The perfused placenta must be maintained in culture medium to ensure its physiological stability and viability. This model is the only available one retaining the whole integrity of the placental structure [121]. As a matter of fact, this configuration allows the intervillous space between the basal decidua (maternal side) and the villi (fetal side) to be kept intact, which offers a large amount of assay possibilities [142].

This model presents the adequate placental barrier’s characteristics to study the placental transfer of xenobiotics and their associated mechanisms [143].

A more recent application of this model was the evaluation of the safety of nanoparticles and nanomedicines. A study about the transplacental passage of 25 and 50 nm silica nanoparticles showed a low permeability towards placenta tissue, but no nanoparticle transfer from the mother side to the fetus side. The complementary use of confocal microscopy showed some accumulation of the silica nanoparticles inside the placental tissues [91].

Myllenen and al. measured the transplacental passage and kinetics of gold nanoparticles of 10, 15, and 30 nm in diameter for six hours of circulation compared to one reference compound, the antipyrine [141] that freely crossed the placenta [144]. The perfusion model showed that none of the particles were found inside the fetal circulation after this time of experiments. Further experiments showed that some of the nanoparticles were trapped inside the placenta.

The dually perfused model has also been investigated to evaluate the transplacental passage of classic drug delivery systems such as dendrimer nanocarriers and liposomes.

Menjoge et al. aimed to study the transplacental transport, kinetics, and biodistribution of fluorescently labeled PAMAM dendrimers across the human-termed placenta using the placenta-perfused ex vivo model. The study was conducted in comparison with antipyrine. Results demonstrated low transplacental transport compared to the reference molecule, with a slight dispersion of the nanocarriers into the placental tissues that was mostly found in the intervillous space and was mostly undetectable in the fetal capillaries. This study highlights that the administration of drugs conjugated to polymers such as dendrimers could limit their transplacental transfer from the maternal to fetal side compared to free small molecules [145].

Bajoria et al. studied the fate of neutral, cationic, and anionic liposomes loaded with carboxyfluorescein, a small hydrophilic fluorescent molecule [146]. They considered the maternal-to-fetal placental transfer and the placental tissue uptake of the free carboxyfluorescein compared to the encapsulated one and thereby evaluated the influence of the surface charge of liposomes on the transport of small hydrophilic molecules across the placenta using perfused placenta. The results showed that anionic and neutral liposomes had a higher transfer from the maternal compartment to the fetal side and a higher placental tissue uptake compared to the control group with the free carboxyfluorescein, whereas the transfer of carboxyfluorescein was prevented by the cationic liposomes and the tissue uptake was like the control group. This information advertised the use liposomes of various surface charge regarding the aim of the therapeutics. The same team conducted another study to evaluate the influence of the lipid composition of cationic liposomes in the maternofetal transfer of warfarin, an anticoagulant molecule [147]. Results highlighted a reduced fetal transfer of the encapsulated drug compared to the free warfarin, with higher placental tissue accumulation of liposomes made with lecithin compared to the ones composed of cholesterol and stearylamine. Cationic liposomes limited the transplacental passage of carboxyfluoscein compared to neural and anionic liposomes. Nevertheless, the conclusions about the transplacental passage of the nanocarrier seems uncertain because the course of nanocarrier was not followed up on; only the carboxyfluorescein was dosed.

Finally, the perfusion model appears as a model of choice to test the transplacental passage of small therapeutic molecules to evaluate the dangerousness towards the fetus in the context of pregnancy. Some works proposed to use it also for safety studies on nanocarriers. However, this model could not provide information about the placental penetration and its consequences for fetus development. For this matter, a complementary ex vivo model is often used as a complementary model to the perfused placenta, which is based on the isolation and culture of entire placental villi, called placental explants.

#### 3.3.2. Human Placental Explants

Human placental explants are an ex vivo model corresponding to the structural and physiological unit of the placenta. This unit is called villi and is isolated from a delivered placenta, and villi are then maintained in culture. It corresponds to a miniature representation of a whole placenta and resembles a 3D co-culture that is already set up. 

Human placental explants consist of small parts of tissue dissected and isolated from either a first-trimester or full-term placenta obtained after birth. This ex vivo model was described in the late 1990s by Sooranna et al. [148]. The protocol to culture it was optimized and enabled the study of various factors, such as physiology, disease processes, toxicology, pharmacology, cellular uptake, etc., with the right culture conditions to get as close as possible to the in-utero situation [149]. As an ex vivo model, placental explants can be maintained in culture up to seven days, which is longer than perfused placenta [150]. Due to these huge advantages, this model can be considered very useful in the study of nanoparticle delivery to the placenta [149,151].

A previous study from our group used human villous placental explants to develop an innovative HPLC method to perform quantitative evaluation of fluorescent lipoplexes upctaken by the placenta. The explant culture was optimized by hanging the villi on a needle and putting them under agitation to recreate the in vivo environment of the floating villi in maternal blood (Figure 5). Results showed that the amount of lipoplexes up taken by the cells increased with the incubation concentration [152]. 

Another study focused on the physiological and structural impact and potential toxicity of dendritic polyglycerol nanocarriers. Nanocarriers of 5 nm with different charges were incubated at different concentrations with first-trimester human placental explants for 6 and 24 h [100]. Results showed a dose-, charge-, and time-dependent accumulation inside the placental tissue. These data suggest that a surface charge should be required to ensure a sufficient uptake of the NPs by the placental cells since the negatively charged NPs were restricted to the outer layer of the placental explants, where the positively charged NPs were found in the mesenchymal axis. This presence inside the mesenchymal axis could indicate a potential crossing of the NPs on the fetal side without any membrane damage or placental suffering. However, transplacental passage should be confirmed on a complementary ex vivo model such as the perfused placenta.

A study conducted by our group assessed the transplacental passage of fluorescently labelled PEGylated neutral liposomes loaded with carboxyfluorescein [151] using those both ex vivo models. Using complementary models and double fluorescent labelling of the drug and the nanocarrier allowed the fate of the carrier and the drug to be followed and precisely understood. The perfused placenta dispensed quantitative data regarding the kinetics and transfer of the drug across the placenta, whereas the explants aimed to understand the uptake of the liposomes inside the placental tissues from a qualitative perspective. Similar transplacental passage rate and placental tissue accumulation were found for the encapsulated and the free compounds. The fluorescence associated with the liposome was accumulated inside the outer cell layer of placental tissues but was not detected inside the fetal circulation. This methodology allows a different behavior between the drug and the carrier to be detected. 

Kaitu’u-Lino et al. evaluated the uptake of doxorubicin nanocell EnGeneIc Delivery Vesicles (EDVs) of 400 nm by the placental explants to treat ectopic pregnancy. Those nanocarriers actively target the placental using anti-EGFR antibodies, which are highly expressed at the surface of the placenta. The purpose of this study was to determine whether these nanovesicles could be internalized by the placental cells to administer chemotherapy to the tumor [153]. Results showed that the functionalized EDVs with the anti-EGFR antibodies were more up taken by the placental explants than the naked EDVs or free doxorubicin. In this study the explant ex vivo model delivered qualitative input about the anatomical localization of the nanovesicles compared to the in vitro model of JEG-3 cells that gave quantitative information about the amount of doxorubicin delivered to the cells. 

Cureton et al. also studied the fate of liposomes with first trimester and term placental explants to deliver a vasodilator specifically to the uterine vessels in the context of impaired uteroplacental perfusion, which is one of the etiologies of fetal growth restriction [46]. In this study, liposomes were decorated with a synthetic peptide sequence that could specifically bound to the endothelium of the uterus arteries in vivo. The functionalized liposomes were incubated with first trimester and term placental explants up to 48 h. Here, the ex vivo explant model was used to determine whether the peptide and peptide-functionalized liposomes could specifically recognize the placenta and be up taken by the placental cells. Results showed cell penetration and accumulation within the syncytiotrophoblast for the first-trimester explants and to a lesser extent for the term ones. The term placental explants were also treated with the therapeutic compound of interest, SE175, a peptide with vasodilatation properties that can be used for fetal growth restriction therapy, and no adverse effects were observed on the endocrine function level, the metabolism, or the structural integrity.

Placental explants arise from the dissection of a single patient’s placenta and most from mothers without any pathology. Therefore, it is possible to have access to placenta from the first trimester and third trimester, which offers a wide variety of placental structures to be studied. It should be highlighted that explants extracted from pathological placenta can exhibit some interesting characteristics that can serve the purpose of nanomedicine studies. For example, in preeclampsia, it has been shown that the placental overexpression of an antiangiogenic soluble protein, sFlT-1, is involved in the physiopathology mechanism. To evaluate new therapeutics against this disease, a model overexpressing this protein appears to be adequate to evaluate the efficiency of the novel drug. One study highlighted that placental explants coming from a mother with preeclampsia secrete in culture medium diverse isoforms of sFlT-1 [154]. This element is important because it shows that not only do placental explants in culture preserve the anatomical integrity of the barrier, but also preserve pathological characteristics.

This versatile model may open a new approach in the study of nanomedicine in a pregnancy context. This innovative model is one of the keys to deeply study the interaction, the penetration/accumulation, and the localization inside the placental tissue of nanocarriers and active pharmaceuticals ingredients.

## 4. Conclusions

Various placenta-based evaluation models have been developed and validated to study nanocarriers and nanoparticles over the years. These models are summarized in Figure 6. 

It appears that cell lines have a real interest concerning the toxicity screening of the nanocarriers of interest. Many studies have been done on the evaluation of transplacental passage and the safety of nanomedicine. For those kinds of studies, in vivo and ex vivo models, especially the dually perfused placenta, appeared as the model of choice. In fact, few studies focused on the evaluation of the interaction and penetration of the nanocarrier into the placenta and in which structure of the placenta. Several studies provide reliable answers concerning the crossing of the placenta towards the fetus or not but do not go deeper into the global nanocarriers’ behavior. It should be mandatory to conduct a study as exhaustive as possible to reassure the safety and the efficacy of the tested product in such a sensitive therapeutic context. That is why the ex vivo model of the placental explants seems to be the most suitable model to use to answer these questions. They tend to be the missing piece in the jigsaw puzzle of the evaluation of nanocarriers during pregnancy. The explant model is essential to evaluate the penetration of nanocarriers inside the placenta to deliver active pharmaceutical ingredients. However, these models could also be used to evaluate the course of nanocarriers that should not interact nor cross the placenta to specifically treat the mother, or nanomedicine that will ensure a full passage through the placenta to specifically treat fetal disorders. It seems natural to use complementary models to cover all aspects of the fate of nanomedicine inside the body of a pregnant women and how efficient it will be. All of this is to create a safe and trustworthy climate to administer medicine to pregnant women in a controlled and secured way.

With this acknowledgment, we propose a flowchart (Figure 7) on how to use and combine various models to evaluate a nanocarrier for a specific application during pregnancy, i.e., all conditions that occur during pregnancy and not only due to pregnancy (i.e., epilepsy, diabetes, asthma, hypertension, preeclampsia, fetal disorders, etc.).

This flowchart proposal intends to open new perspectives in placental models’ utilization for nanocarrier evaluation, hoping to soon achieve a standardized methodology that will allow for a better understanding of the behavior of nanocarriers with placenta and for a relevant comparison between studies realized in different labs to be established.

The evaluation of nanocarriers is dependent of the application: whether it is to treat the mother or the placenta with no transplacental passage, or whether it is to treat the fetus in utero. For each application, there are different characteristics that should be investigated, and therefore several model associations to answer these questions. Globally, to screen various formulations it seems suitable to use a combination of in vitro and ex vivo models because they present complementary characteristics: For example, the use of cell culture can give useful information about the toxicity, whereas the use of the perfused placenta or a cell culture on Transwell^®^ can inform about the transplacental passage of nanocarriers. Therefore, we suggest the ex vivo explant model to complete the toxicity assay concerning a putative placental accumulation and damage of nanocarriers and/or a loaded drug.

The study of nanomedicine requires the study of specific parameters compared to classic drug dosage forms, especially for pregnancy-associated disorders. In this case, the most important question concerns the safety of both the mother and the baby, which implies studying the transplacental passage of drugs. It has been shown that behavior of APIs towards the placenta depends on their physicochemical characteristics. The encapsulation of APIs inside a nanocarrier can increase or reduce the transplacental passage of APIs regarding the targeted application. Therefore, it seems there is a need for a case-by-case assessment to develop new therapeutics for pregnant women based on nanomedicine and to choose wisely and purposely the right models to include in the study according to the objective. 

With the rising interest in nanomedicine for pregnancy therapeutics, we noticed a rising number of scientific articles about this topic over the past few decades. We also found that innovative placental models have emerged, such as organoids and placenta-on-a-chip, which could provide relevant information in controlled experimental conditions but have not been yet used to evaluate nanocarriers. It will be interesting to conduct a new review study once more papers are published on this topic. However, in the end, despite the low number of papers included, this review allowed us to summarize and categorize available placental models to study nanocarriers for pregnancy-associated disorder care and to highlight general trends of nanoparticle interactions with placenta.

## Figures and Tables

**Figure 1 biomedicines-10-00936-f001:**
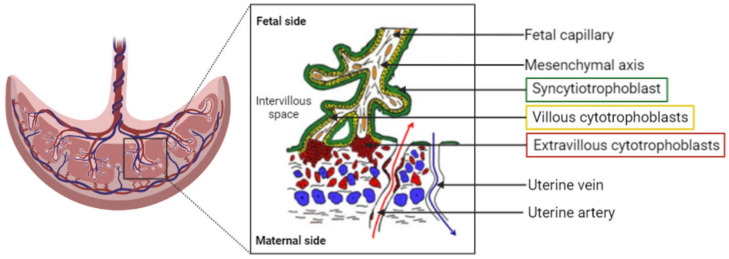
Representation of the organization of the human placenta, maternal–fetal blood circulation, and the villous, the placenta functional and structural unit. A villous is composed of a mesenchymal axis (white) comprising fetal capillaries (orange) grouped together to form the umbilical cord. The mesenchymal axis is covered by a specific cell type: the villous cytotrophoblasts (yellow), which fuse to renew the syncytiotrophoblast (green). The syncytiotrophoblast is in direct contact with the maternal blood within the intervillous space/chamber. The oxygenated blood is flooding the placenta from the uterine arteries (red arrow), whereas deoxygenated blood exits the intervillous space through the uterine veins (blue arrow).

**Figure 2 biomedicines-10-00936-f002:**
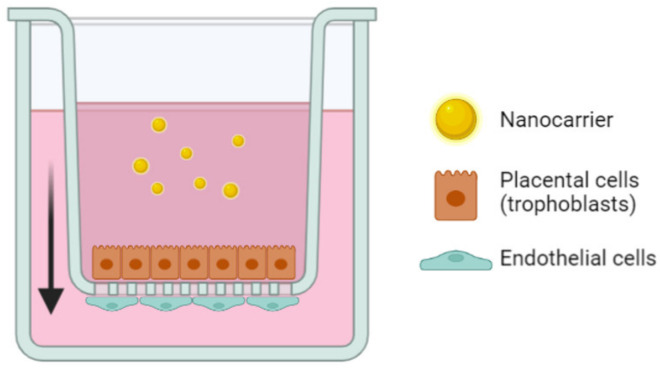
Representation of the Transwell^®^ setup to recreate the placental barrier in vitro using trophoblasts and endothelial cells to assess the transplacental passage of nanocarriers. The arrow represents the transepithelial direction of the nanoparticles in the model.

**Figure 3 biomedicines-10-00936-f003:**
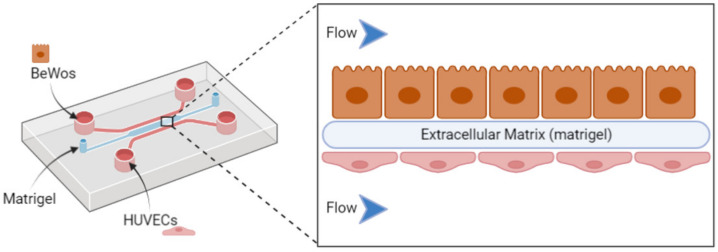
Representation of a placenta-on-a-chip setup using a microfluidic chip, including inlets of trophoblast cell lines (BeWo) and placental endothelial cells (HUVEC) to evaluate nanocarrier behavior with the placental barrier.

**Figure 4 biomedicines-10-00936-f004:**
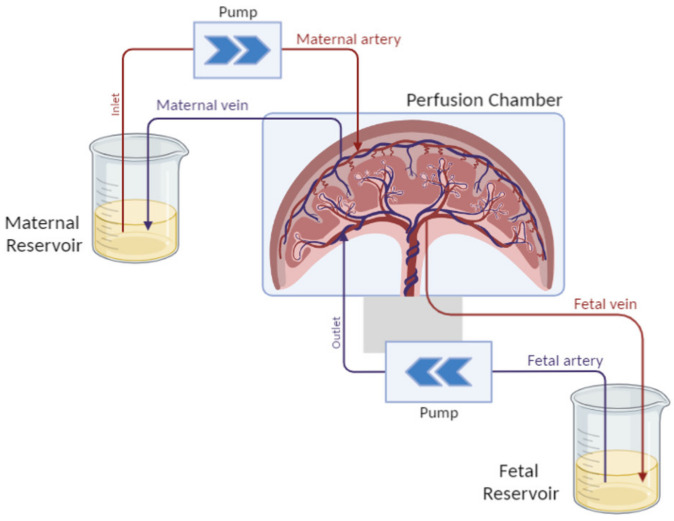
Schematic representation of the dually perfused placenta setup. The placenta is maintained in culture conditions in a perfusion chamber and is perfused by catheters on both sides, each one connected to a specific compartment mimicking the maternal blood flowing into the placenta to reach the fetal blood circulation.

**Figure 5 biomedicines-10-00936-f005:**
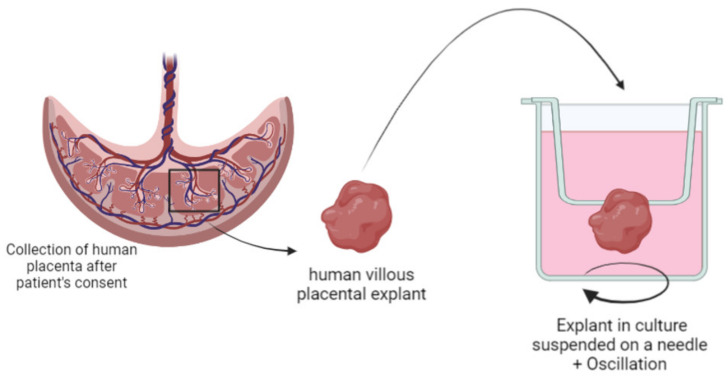
Schematic representation of villous placental explant isolation and specific culture conditions of “floating villi” hanging on a needle.

**Figure 6 biomedicines-10-00936-f006:**
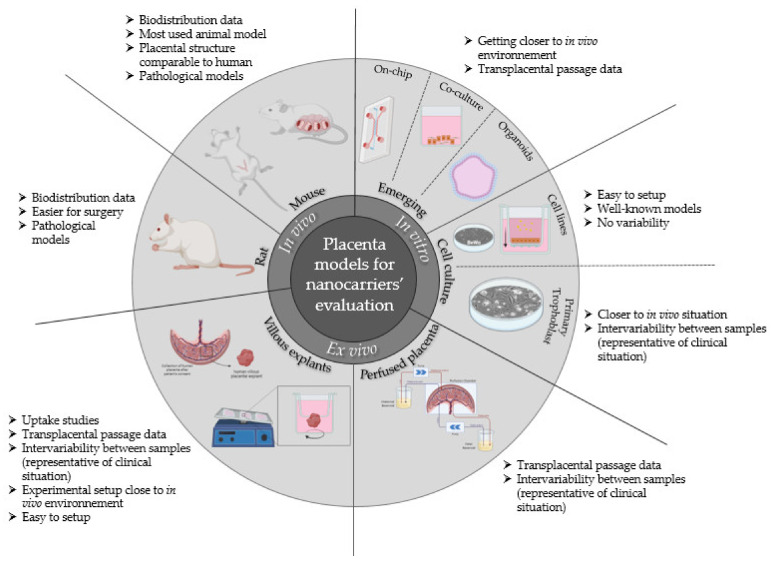
Summary of preclinical placenta-based evaluation models available for nanocarrier and nanoparticle evaluation on the placental barrier for therapeutic management of pregnancy-associated disorders and relevant information coming from their use.

**Figure 7 biomedicines-10-00936-f007:**
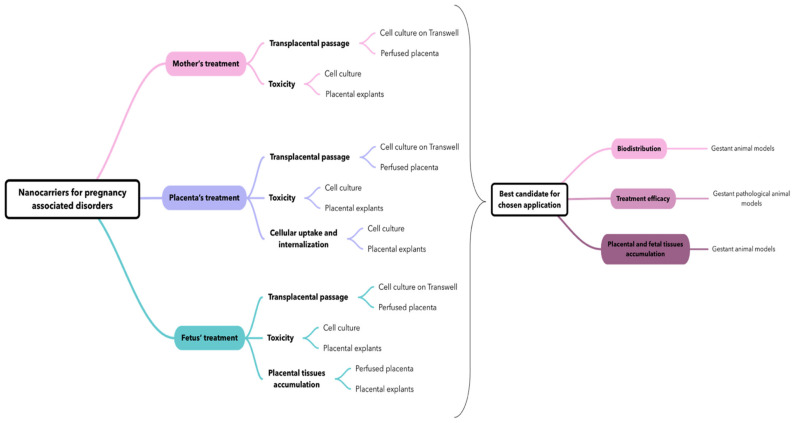
Flowchart on the possible usage and combination of placenta-based models to study nanocarriers for care management of pregnancy-associated disorders.

**Table 1 biomedicines-10-00936-t001:** Anatomical and physiological comparison of placenta between rodents and human primates.

Characteristics		Rodents	Human Primates
Anatomy [60]	Morphology	Discoid: a single placenta is formed in a discoid shape (Villi are connected in a disk)	
	Structure [61]	Labyrinthine placenta	Villous placenta
	Histology	Hemochorial placenta: Placental tissues are bathed in maternal blood (opposition to other placenta where fetal tissues are separated from maternal blood by 2 or 3 layers of cells).	
		Hemotrichorial: one layer of cytotrophoblast based on two layers of basal syncytiotrophoblast	Hemodichorial: one layer of syncytiotrophoblast upon one layer of cytotrophoblast on basal layer
Physiological functions—major differences [62]		Progesterone production provided by corpus luteum indispensable during the whole pregnancy. Chorionic gonadotropin hormone presence has not been demonstrated	Progesterone production provided by corpus luteum and after placental production takes over gradually. Chorionic gonadotropin hormone presence

**Table 2 biomedicines-10-00936-t002:** Summary of the main cultured cell models used in in vitro experiments on placenta.

Immortalized Human Trophoblasts	Trophoblasts Derived from Human Choriocarcinoma	Hybrid Cell Line
HTR-8/Svneo—1st T (SV 40)	BeWo	ACH-3P
Swan 71—1st T (SV 40)	JEG-3	
	JAR	

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
