# Peer review of "Placental Models for Evaluation of Nanocarriers as Drug Delivery Systems for Pregnancy Associated Disorders"

_biomedicines, 2022, doi:10.3390/biomedicines10050936_

Round 1

Reviewer 1 Report

This is a narrative review that aims to describe the various placenta-based models suitable for nanocarriers evaluation for pregnancy associated therapies.

There has been a lot of work for this paper.

There are several issues to be revisited before consideration of this article as suitable for publication in the journal.

Language and syntax should be revisited throughout the text.

The introduction section should contain more information on the gap in the literature on the specific subject of the review, leading to the subsequent rationale of the study.

I would urge authors to add the necessary sections of materials and methods, search algorithm, and synchronization of the sections among them. This would guide the reader in a more friendly way, when reading the paper.

The same observation/suggestion stands for results, discussion, where the limitations of this review is missing, along with the future proposals for further research.

Author Response

We would like to thank the reviewer for the constrictive comments, please find below our answers point by point. All modifications in the text have been highlighted in yellow.

This is a narrative review that aims to describe the various placenta-based models suitable for nanocarriers evaluation for pregnancy associated therapies.

There has been a lot of work for this paper.

There are several issues to be revisited before consideration of this article as suitable for publication in the journal.

  • Language and syntax should be revisited throughout the text.

Thank you for this comment.
The whole manuscript has been checked again by the authors and we revised the syntax as much as we could.

  • The introduction section should contain more information on the gap in the literature on the specific subject of the review, leading to the subsequent rationale of the study.

Thank very much for this very relevant input.

Some paragraphs have been added to the manuscript to answer this comment.

Lines 83-88:

 « Finally, among scientific publications analyzed, we can highlight the absence of a consensus on which model of the placental barrier to use to evaluate a specific property of a nanocarrier. Therefore, we propose a flow chart on how to use placenta-based models to screen and to help to build a homogenized work package to evaluate nanocarriers potential for pregnancy associated disorders. »

Lines 837-840:

 “This flowchart proposal intends to open new perspectives in placental models’ utilization for nanocarriers evaluation, hoping to achieve, soon, a standardized methodology that will allow to better understate of the behavior of nanocarriers with placenta and to establish a relevant comparison between studies realized in different labs.

  • I would urge authors to add the necessary sections of materials and methods, search algorithm, and synchronization of the sections among them. This would guide the reader in a more friendly way, when reading the paper.

Thank you for this comment.

As a matter of fact, we didn’t consider this review as a meta-analysis. We analyzed papers dealing with the study of nanocarriers as drug delivery systems and their evaluation with the placental barrier. In this scope we decided not to include papers studying the toxicity of environmental nanoparticles or endocrine disruptors. Therefore, the number of papers included in each section was too low to perform such a statistical analysis and we think divided each part of the review with material and methods, search algorithm and results might not be applicable for this comprehensive review.

We added a paragraph in the introduction to justify the choice of papers included in this review.

Lines 67-74:
 “In this review, we analyzed published articles reporting the evaluation of nanocarriers as drug delivery systems for pregnancy associated disorders. Therefore, studies discussing the toxicity or impact of environmental particles on pregnancy outcomes were not included in this review. The aim of the current study is to review the various models which could be used to evaluate nanocarriers interaction with the placental barrier for novel therapeutics development for pregnant women. For easier understanding, nanoparticles referring to model nanoparticles used for fundamental studies will be designated as nanoparticles (NPs) and nanoparticles studied as drug delivery systems will be named nanocarriers.”

Finally, we added tables to summarize the papers included at the end of each section and to highlight the relevant information to be retained in each study. 

  • The same observation/suggestion stands for results, discussion, where the limitation of this review is missing, along with the future proposals for further research.

Thank you for this insightful comment.

We added a paragraph the discuss the main limitation of the review, which is the small numbers of papers included, and we propose to conduct a new review in the future, once more papers will be published, especially when studies will be conducted using innovative placental models such as organoids or placenta on-a-chip.

Lines 861-869:

 “With the rising interest of nanomedicine for pregnancy therapeutics, we could notice a rising number of scientific articles about this topic over the past decades. We also found that innovative placental models emerge such as, organoids or placenta on-a-chip, which could provide relevant information in a controlled experimental conditions but haven’t been yet used to evaluate nanocarriers. It will be interesting to conduct a new review study once more papers will be published on this topic. But in the end, despite the low number of papers included, this review allowed to summarize and categorize available placental models to study nanocarriers for pregnancy associated disorders care and to highlight general trends of nanoparticles interactions with placenta.”

Reviewer 2 Report

 In this review article the authors summarized the recent progress on nanocarriers as drug delivery systems for pregnancy-associated diseases. The manuscript is interesting and basically well written, thus, the reviewer has only minor comments for revision.

Minor comments.

1, Fig.7: The size of the letters appears too small to read.

2, Several references are not cited yet. (Line 251, 287, 355 etc.)

3, Some words appear to be mistyped. (line 644, 722 etc.) Please check whole manuscripts.

Author Response

Authors thanks the reviewer for the positive evaluation. Please find below our answer point by point. All modifications in the text have been highlighted in yellow.

 In this review article the authors summarized the recent progress on nanocarriers as drug delivery systems for pregnancy-associated diseases. The manuscript is interesting and basically well written, thus, the reviewer has only minor comments for revision.

Minor comments.

  • 7: The size of the letters appears too small to read.

We are sorry for this inconvenience, thank you for this remark.
The figure has been enlarged and inserted in landscape mode so the reading can be facilitated.

  • Several references are not cited yet. (Line 251, 287, 355 etc.)

Thanks for the comment. References have been added regarding the requirements.

Line 275: (Cross, 2016; Rinkenberger and Werb, 2000; Wooding and Burton, 2008)

Line 318: (Orendi et al., 2011)

Line 384: (Rytting et al., 2012)

Line 616 : (Aengenheister et al., 2021; Wick et al., 2010)

  • Some words appear to be mistyped. (line 644, 722 etc.) Please check whole manuscripts.

Thank you for this comment.
The whole manuscript has been checked again by the author and we managed to correct mistyped words.

Round 2

Reviewer 1 Report

I think that the paper has been improved.